# Safety-Promoting Interventions for the Older Person with Hip Fracture on Returning Home: A Protocol for a Systematic Review

**DOI:** 10.3390/jpm12050654

**Published:** 2022-04-19

**Authors:** Paula Rocha, Cristina Lavareda Baixinho, Andréa Marques, Adriana Henriques

**Affiliations:** 1Local Health Unit of Guarda, 6300-749 Guarda, Portugal; 2PhD Student of University of Lisbon, 1649-004 Lisboa, Portugal; 3Nursing Research, Innovation and Development Centre of Lisbon (CIDNUR), 1900-160 Lisboa, Portugal; crbaixinho@esel.pt (C.L.B.); ahenriques@esel.pt (A.H.); 4Center for Innovative Care and Health Technology (ciTechCare), Polytechnic of Leiria, 2411-901 Leiria, Portugal; 5Nursing School of Lisbon, 1900-160 Lisboa, Portugal; 6Nursing School of Coimbra, Health Sciences Research Unit: Nursing (UICISA:E), 3000-232 Coimbra, Portugal; andreamarques23@esenfc.pt; 7Department of Rheumatology, Centro Hospitalar e Universitário de Coimbra, 3000-071 Coimbra, Portugal

**Keywords:** hip fracture, hospital discharge, older person, returning home, safety

## Abstract

Ageing and physical frailty associated with decrease in muscle and bone mass lead to the older persons’ vulnerability and increased risk of falling. It is estimated that one in every ten falls in this age group results in a fracture, leading to a downward spiral in their health status, causing greater dependence, with a progressive functional decline that makes it difficult to return to their functional and social status prior to the fracture. The aim of this study is to identify the available evidence on the interventions that promote the safety of older people with hip fracture after hospital discharge. A search will be performed in MEDLINE and CINAHL databases. Randomised and controlled studies that focus on functional assessment, performance in activities of daily living, level of concern about falls, risk and prevalence of falls, injuries secondary to falls, re-fracture rate and health-related quality of life in hip fracture patients will be included. Two authors will perform the study selection, data extraction, and quality assessment independently. Any disagreements will be resolved through discussion with a third researcher. Methodological quality of the included trials will be evaluated by the Cochrane risk-of-bias criteria, and the Standards for Reporting Interventions in Controlled Trials.

## 1. Introduction

Fractures resulting from falls trigger a downward spiral in the health status of older persons, causing greater dependence and disability, and may lead to long-term complications [1]. This condition is also associated with an increase in mortality when compared to the general population, a marked decline in quality of life, and consequently represents a notable burden on health systems [2].

Osteoporosis, characterized by a reduction in bone mineral density and micro-structural deterioration of bone tissue, increases bone fragility, inducing greater susceptibility to fragility fractures [3,4,5]. With the increase in life expectancy and the ageing of the population, the occurrence of fragility fractures, and more specifically of hip fractures (HF), has been increasing [2,4]. Most fragility fractures occur in people aged over 65 years, and result from low-impact trauma to osteoporotic and, therefore, more fragile bone, often associated with a fall [6,7]. Women are the most affected by this situation, at a ratio of 3:1 compared to man [2].

This clinical condition, more prevalent in older people [4], is asymptomatic until the moment when the first manifestation associated with a low impact trauma appears, often causing a fragility fracture [8], so reducing the risk of falls is crucial. Approximately 1/3 of the elderly aged 65 years suffer at least one fall per year; above 85 years this risk increases to 50% [9,10]. The fracture that occurs in 5–10% of fall events leads to difficulty in returning to the person’s functional and social state prior to the fracture [11]. This leads to a need for care, with the main goal being unequivocally to maximize functional potential and provide the quickest possible return to the previous level of functionality [12]. This objective is achieved not only through surgical intervention, which provides the necessary stability to allow early mobilization and locomotion, but also through patient rehabilitation in the post-surgical context, with the aim of maximizing functional potential [12].

The significant impact that these fractures have in terms of public health is related to their high prevalence, the medical consequences that they entail, and also the reduction in the quality of life of the patient and their caregivers [6]. In this respect, it is estimated that 20–30% of patients with these fractures die in the following year; 50–60% of cases present some type of functional and/or motor loss, and only 30–40% of patients achieve a level of functional recovery similar to that before the HF [13]. Most of these patients require continued care for a long period, and 20% of them are institutionalised during the year following the fracture, which is the main reason for the loss of self-care in carrying out the activities of daily living [10,14]. 

Several studies have shown the importance of transitional care in the process of rehabilitation and readaptation of the elderly person during the process of discharge from hospital to home [15,16,17]. Transitional care should start at the time of admission, continue during hospitalisation and remain after discharge, which implies changing the ways of providing continuity of care to these people, since hospital environments only address biological aspects, and are not operationally designed to help older people recover or improve their functionality and return to their activities after hospitalisation [17,18].

In a study analysing the transition care of elderly patients with hip fracture, eight relevant domains were identified, including the complexity of the patient and system constraints, which, as inherent factors to the context, tend to hinder transition care. Others include patient involvement and choice, the role of the family caregiver, strong relationships, role coordination, documentation and information sharing are areas with the potential to support and improve transition care [19].

Involving patients in their care process by encouraging their participation may result in better outcomes [20]. The implementation of interventions that integrate exercises should include strategies to motivate and adhere to exercise, as well as contents related to components of the home care programme [21].

In view of the above, the aim of this systematic review is to identify the available evidence on interventions that promote the safety of older people with hip fracture after hospital discharge.

## 2. Materials and Methods

### 2.1. Type of Study

The protocol for this systematic review was designed according to the recommendations of the Preferred Reporting Items for Systematic Review and Meta-Analysis (PRISMA) Protocols checklist [22], PRISMA-P (Preferred Reporting Items for Systematic review and Meta-Analysis Protocols) [23] and the Cochrane Guidelines for Systematic Reviews of Interventions [24].

To answer the research question, which interventions promote the safety of older people returning home after a HF? the option was the use of a systematic method that enables reliable results to be obtained, from which conclusions can be drawn and decisions made, minimizing the risk of bias [22,24], and to guide clinics and health policies on the basis of research results.

### 2.2. Eligibility Criteria

All randomised controlled trials (RCT’s) addressing interventions promoting the safety of the elderly person with proximal femur fracture in returning home will be included.

The target population will be people with hip fracture (including trochanteric, neck and subtrochanteric fractures) aged over 65 years, who had been admitted to hospital and had undergone orthopaedic surgery.

The types of interventions that promote the safety of the elderly person with HF, after hospital discharge, in returning home, are described below and grouped into categories that emerged from the literature review. As this is an incipient field of global research, this list is not exhaustive, providing only a few examples of the diverse interventions that have been carried out:-Health literacy activities targeting the patient and caregiver to reduce complications associated with fracture.-Follow-up programmes and monitoring of the functional assessment and performance in the activities of daily living.-Programmes to reduce/control the risk and prevalence of falls and/or the occurrence of new falls.-Interventions to monitor and improve health-related quality of life after HF.-Environmental modification interventions to promote accessibility and mobility indoors.-Transitional care interventions to promote adherence to the rehabilitation programme upon return home.-Action plans that include strategies to improve the quality of care and follow-up of older people with fractures on their return home [1,7,11,12,15,16,17,18,19,20,21,25].

Exclusion criteria will be established for studies addressing interventions related to conventional treatment, considering a conventional treatment as that in which the person does not receive any additional intervention beyond verbal advice and/or medication optimisation, and interventions for the adult population with fragility fractures.

The review enables the identification and organisation of interventions with evidence for promoting safety after hospital discharge.

### 2.3. Research Strategy

Taking into account the nature of the research carried out, the strategy adopted will consist of conducting a bibliographical search in the following electronic databases: MEDLINE, and CINAHL. A list of references of the previously selected relevant articles will be compiled, and a search made for possible additional studies that may be relevant, with the aim of increasing the sensitivity of the research.

The search strategy will be developed using the medical subject headings (MeSH) and, in line with them, was based on the following key concepts: “Safety”, “Functional Independence”, “Elderly”, “Hip fracture”. The search in the CINAHL database will use the Subject Headings.

The search strategy adopted according to the respective database (Table 1) will be limited to the last 10 years and to the English and Portuguese languages.

### 2.4. Selection Process

Two reviewers will independently select, extract and code the data. In turn, a third reviewer will mediate any discrepancies in the selection of studies [22,24].

The results of the research will be inserted in the Rayyan^®^ platform, facilitating collaboration between the three reviewers in the process of study selection.

Thus, all the titles and abstracts obtained in the research will be analysed in relation to the established inclusion criteria, with the aim of assessing the eligibility of the studies. Studies considered relevant will be selected for further reading and then included in the review. The reasons for exclusion of the studies will be documented.

For each of the studies that fulfil the inclusion criteria, data extraction and codification will be carried out by two reviewers (independently), using a previously prepared, simple and standardised form that describes the characteristics of the study. The data extraction form will include:Study design, title, authors’ names, publication date, country, sample size, funding source.Characteristics of the population (number of participants, average age, proportion of each sex).Method (sampling, instruments and randomisation method).Characteristics of the intervention (type and period of intervention).Results.

### 2.5. Assessment of Methodological Quality and Risk of Bias

Two reviewers will perform the quality assessment of the included studies independently. A third reviewer may clarify any discrepancy that may arise. If the reviewers are co-authors of some studies, they will not evaluate the risk of bias in these studies [22].

The assessment of the risk of bias of the included studies will take into account the potential sources of bias and may determine a low risk of bias (if all criteria were fulfilled), moderate risk of bias (one or more criteria were partially fulfilled), high risk of bias (one or more criteria were not fulfilled) and unclear (not enough information is available to assess the study with regard to the risk of bias). The Cochrane Risk of Bias Assessment Tool [22,24] will be applied.

### 2.6. Data Synthesis

The synthesis of the information regarding the characteristics and results of the included studies will be presented in a table and complemented with a narrative summary, which will assess the methods used, and the results of the studies.

If there are sufficient RCTs, a meta-analysis will be produced, in which synthesis of the data will be carried out using the Review Manager software^®^ (version 5.3, Cochrane Collaboration, London, UK). If it is not possible to perform meta-analysis, the results will be presented descriptively [22]. The strength of the body of evidence will be assessed using the GRADE framework [22,23,24].

The results of the systematic review will be reported according to the PRISMA guidelines [22].

## 3. Discussion

It is estimated that around 50–60% of people that have suffered a HF have some type of functional and/or motor loss, and that in only 30–40% of cases is a level of functional recovery attained similar to that which existed before the fracture, thus requiring continued care for a long period of time to reduce the loss of independence to perform activities of daily living [13,17].

On returning home, associated with the restriction of self-imposed and hetero-imposed mobility and loss of gait capacity, these people frequently present a functional decline, unable to recover to pre-fracture functional levels. Sometimes, it is also associated with an increased risk of falling and difficulties in accessing healthcare [11,13,26,27,28], which, together, represent serious problems, with a great impact on people and health services.

After the occurrence of a fragility fracture there is a high risk of a subsequent one, so the prevention of a new fall is a key aspect, requiring the development of strategies to increase safety at home [17]. The development of post-discharge transition programmes has proved to be successful in contributing to reduce readmissions and disability [25]. However, there is no systematisation of many of the interventions implemented in these programmes, nor indication of their effectiveness in maintaining the person’s safety in his/her home environment, enabling recovery and avoiding postoperative complications.

Other authors have observed that despite the growing increase in scientific knowledge, there have been some gaps in the description of interventions, which have consequently made the assessment of results and their replication and implementation in practice difficult [7,21], with an important impact on the process of transition from hospital to home.

After the occurrence of this type of fracture, rehabilitation carried out in the post-acute phase aims to rescue as much autonomy as possible for patients affected by such pathological situations; however, its duration is sometimes clearly insufficient to ensure an effective and lasting result on the return home [11]. In this context, home-based rehabilitation is recommended, producing considerable positive effects on physical functioning after a femur fracture, and contributing towards a significant improvement in mobility, daily activity, instrumental activity and balance [8].

Thus, this systematic review will strengthen the evidence base on safety-promoting interventions for older people with proximal femur fracture upon return home, exploring potential outcomes regarding functional assessment, performance in activities of daily living, level of concern about falls, risk and prevalence of falls, injuries secondary to falls, fracture rate, and health-related quality of life. It could potentially benefit health professionals and researchers by facilitating the design and implementation of interventions to improve the care of these people. It will also provide information that may make an important contribution to policymaking regarding the development of strategies to improve the quality of care and follow-up of older people after HF.

This review has potential limitations related to the fact that the search strategy did not include sources of information available in languages other than English and Portuguese.

## 4. Conclusions

This systematic review protocol will allow identification of the available evidence on the interventions that promote the safety of the elderly person with proximal femur fracture upon returning home, thus contributing to the systematisation of transitional care interventions that promote rehabilitation, prevent the recurrence of falls, the immobility syndrome and other postoperative complications associated with this type of surgery, which translate into loss of quality of life and decreased life expectancy.

The systematisation of these interventions will make it possible to guide the clinic and contribute to the training of health professionals in this area.

## Figures and Tables

**Table 1 jpm-12-00654-t001:** Search Strategy. Lisbon, 2022.

	Search Strategy	Number of Articles
#1	Elderl*”“[Title/Abstract] OR ““aged”“[Title/Abstract] OR Age* *”“[Title/Abstract] OR Older Person *”“[Title/Abstract] OR Older Adult *”“[Title/Abstract] OR ((““Aged”“[Mesh]) OR ““Frail Elderly”“[Mesh])) OR Frail Older Adults ““[Mesh] OR Frail Older Adult ““[Mesh] NOT ““animals”“[Mesh]))	395,080
#2	hip fracture*”“[Title/Abstract] OR ““fragility”“[Title/Abstract] OR ““fragility fracture”“[Title/Abstract] OR ““osteoporos*”“[Title/Abstract]) OR (““Hip Fractures”“[Mesh])) OR Hip injuries [Title/Abstract]) OR hip joint [Title/Abstract])(((((““Osteoporosis”“[Mesh])) OR ““Osteoporotic Fractures”“[Mesh])	45,597
#3	(““Rehabilitation”“[Title/Abstract] OR ““Rehabilitation exercise”““[Title/Abstract] “OR(““Rehabilitation program ““[Title/Abstract] OR”(““Rehabilitation intervention “[Title/Abstract] OR ““Education program*”“[Title/Abstract] OR ““Educational programme*”“ ““Education intervention”“[Title/Abstract] OR ““Educational strateg*”“[Title/Abstract] OR ““Educational tool*”“[Title/Abstract] OR ““Healthy-interactions”“[Title/Abstract] OR ““Educational therapy”“[Title/Abstract] OR ““Health literature”“[Title/Abstract] OR ““Activities of daily living programs”“[Title/Abstract] OR ““Safety promot*”“[Title/Abstract] OR ““Patient education”“[Title/Abstract] OR ““Program*”“[Title/Abstract] OR ““Tool*”“[Title/Abstract] OR ““training”“[Title/Abstract]) OR (((((((““Rehabilitation”“[Mesh]) OR ““rehabilitation”“ [Subheading]) OR ““Rehabilitation Nursing”“[Mesh]) OR ““Orthopedic Procedures”“[Mesh]) OR ““Exercise Therapy”“[Mesh])) OR (““Patient Education as Topic”“[Mesh])) OR rehabilitation/education ““[Mesh]) OR ““mobilization”“[Title/Abstract] OR ““Early Ambulation”“[Mesh])) OR ““exercise training”“[Title/Abstract] OR ““exercise”“[Mesh])) ““gait training”“[Title/Abstract] OR ““Walking Speed”“[Mesh])) OR ““ environmental risk control”“[Title/Abstract] OR ““ daily living activity”“[Title/Abstract] OR ““Activities of Daily Living”“[Mesh]))	116,945
#4	““risk of falling”“[Title/Abstract] OR ““prevalence of falls”“[Title/Abstract] OR ““re-fracture”“[Title/Abstract] OR ““Injuries secondary to the fall”“[Title/Abstract] OR ““quality of life”“[Title/Abstract] OR ““health-related quality of life”“[Title/Abstract] OR ““security”“[Title/Abstract] OR ““functional independence”“[Title/Abstract]) OR Readmissions ““[Title/Abstract] OR Gait Ability OR (““Quality of Life”“[Mesh])) OR (““Quality Indicators, Health Care”“[Mesh])) OR (““Functional Status”“[Mesh])))	494,110
#5	((““randomized controlled trial”“[Title/Abstract] OR ““controlled clinical trial”“[Title/Abstract] OR ““randomized”“[Title/Abstract]) OR (““Randomized Controlled Trial”“ [Publication Type]))”	861,759
#1 AND #2 AND #3 AND #4 AND #5		4860

## Data Availability

Data are available only upon request to the authors.

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
