# Peer review of "Safety-Promoting Interventions for the Older Person with Hip Fracture on Returning Home: A Protocol for a Systematic Review"

_jpm, 2022, doi:10.3390/jpm12050654_

Round 1
Reviewer 1 Report
this paper need a major revision. As written in title 'a Protocol for a Systematic Review' i can not find the protocol.
this paper need major revision because does not exist a correlation between title and content.
The Material and method are general informations without a protocol as suggested in the title.
The authors are presenting some searching criterias without a correlation with the falling prevention or recovery after hip surgery.
Author Response
We thank you for your review of our protocol proposal. We have reviewed it and introduced changes that are shaded in yellow.
Attached to this response is the PRISMA-P checklist with identification of each item in our text.

Reviewer 2 Report
This manuscript presented the good topic and need to be deep design and study. After well explanations the study methodology and design but can not build the final conclusions through the solid results. We need more evidence to improve it can be worked.
Author Response

(The authors gave the same response as above.)

Round 2
Reviewer 1 Report
You have solve the problems of this article and from my point of view it can be published.
Reviewer 2 Report
This manuscript was presented with good study methodology and design but still showed no innovation conclusions through the results from the system reviews.